# Environmental Air Pollutants Affecting Skin Functions with Systemic Implications

**DOI:** 10.3390/ijms241310502

**Published:** 2023-06-22

**Authors:** Georgeta Bocheva, Radomir M. Slominski, Andrzej T. Slominski

**Affiliations:** 1Department of Pharmacology and Toxicology, Medical University of Sofia, 1431 Sofia, Bulgaria; bocheva_georgeta@yahoo.com; 2Department of Genetics, Informatics Institute in the School of Medicine, University of Alabama at Birmingham, Birmingham, AL 35294, USA; rslominski@uabmc.edu; 3Department of Dermatology, Cancer Chemoprevention Program, Comprehensive Cancer Center, University of Alabama at Birmingham, Birmingham, AL 35294, USA; 4Veteran Administration Medical Center, Birmingham, AL 35294, USA

**Keywords:** air pollutants, photopollution, cigarette smoke, AhR, skin diseases

## Abstract

The increase in air pollution worldwide represents an environmental risk factor that has global implications for the health of humans worldwide. The skin of billions of people is exposed to a mixture of harmful air pollutants, which can affect its physiology and are responsible for cutaneous damage. Some polycyclic aromatic hydrocarbons are photoreactive and could be activated by ultraviolet radiation (UVR). Therefore, such UVR exposure would enhance their deleterious effects on the skin. Air pollution also affects vitamin D synthesis by reducing UVB radiation, which is essential for the production of vitamin D_3_, tachysterol, and lumisterol derivatives. Ambient air pollutants, photopollution, blue-light pollution, and cigarette smoke compromise cutaneous structural integrity, can interact with human skin microbiota, and trigger or exacerbate a range of skin diseases through various mechanisms. Generally, air pollution elicits an oxidative stress response on the skin that can activate the inflammatory responses. The aryl hydrocarbon receptor (AhR) can act as a sensor for small molecules such as air pollutants and plays a crucial role in responses to (photo)pollution. On the other hand, targeting AhR/Nrf2 is emerging as a novel treatment option for air pollutants that induce or exacerbate inflammatory skin diseases. Therefore, AhR with downstream regulatory pathways would represent a crucial signaling system regulating the skin phenotype in a Yin and Yang fashion defined by the chemical nature of the activating factor and the cellular and tissue context.

## 1. Introduction

Environmental pollution is a global health problem associated with growing respiratory [1], cardiovascular [2], cutaneous [3,4], and neurological morbidities [5,6], as well as premature mortality [7,8]. Airborne pollution is defined as contamination by any chemical, physical, or biological agent from biogenic, anthropogenic, or mixed origins that modifies the natural characteristic of the atmosphere. Environmental exposure throughout one’s lifespan may lead to epigenetic changes that vary significantly with different types of air pollutants [9]. The exposure time window is also of great relevance for more severe outcomes at certain ages (e.g., childhood and advanced age). Moreover, air pollution exposure during pregnancy (in utero) may cause reduced fetal growth, pre-term birth, or spontaneous abortions [10]. As a first-line barrier, the human skin, together with the lungs, is the main target organ affected by the harmful insults of air pollution [11]. Billions of people are still exposed to unhealthy ambient air, of which the mixture of pollutants could affect skin physiology either directly by transcutaneous absorption or indirectly by their systemic distribution after inhalation or ingestion. The air pollutants of global concern include particulate matter (PM), polycyclic aromatic hydrocarbons (PAHs), benzene, ground-level ozone (O_3_) and other gaseous pollutants such as carbon monoxide (CO), nitrogen dioxide (NO_2_), and sulfur dioxide (SO_2_), various heavy metals, and volatile organic compounds (VOC).

PM is among the most harmful pollutants, being a heterogeneous mixture of very small particles and liquid droplets composed of organic chemicals, acids, metals, and soil or dust particles. Based on their size, they are subdivided into coarse particles with a diameter of 2.5 to 10 μm (PM_10_), fine particles with less than or equal to 2.5 μm (PM_2.5_), and ultrafine particular matter (UFPM) with a diameter under 100 nm (PM_0.1_), also called nanoparticles [12]. Currently, UFPM is considered the most abundant PM pollutant in the atmosphere of industrial and urban areas. The ambient PM through adhesion to the skin surface can reduce epidermal barrier function [13]. The transdermal penetration of solid particles depends on their size and shape, as well as on the state of the skin barrier. The nanoparticles that are larger than 45 nm cannot penetrate the skin. UFPM with a size between 21 and 45 nm can only penetrate and permeate skin with an impaired barrier in comparison with the smaller 4–20 nm nanoparticles, which can potentially permeate not only damaged skin but intact skin as well. UFPM equal to or smaller than 4 nm can penetrate intact skin [14]. Fine and ultrafine-sized particles (PM_2.5_ and PM_0.1_) are respirable and could affect pulmonary function [15]. Additionally, they can potentially reach the alveoli and cross the alveolar-capillary barrier of the lungs, gaining access via blood circulation to various organs [16,17]. In fact, UFPM has a large surface area and the capacity to adsorb PAHs, thus they can reach systemic blood circulation and deep skin, either from lung alveoli or via the skin surface, hair follicles, or eccrine and apocrine glands. Airborne PAHs are widespread and can be found in all human body fluids and hair [18]. PAHs are highly lipophilic, which allows them to easily penetrate the epidermal skin barrier. Moreover, various PAHs are present in cigarette smoke [19], leading to the accumulation of PAH-metabolites in the hair of smokers compared to non-smokers [20]. In fact, cigarette smoke contains several PAHs similar to those defined as air pollutants. Many PAHs can exert mutagenic and cytotoxic effects, and some of them could become phototoxic. Photoactivation of PAH upon ultraviolet A (UVA) light exposure would augment the deleterious effects of PM on the skin [21]. In Europe, PAH levels in the air are estimated via monitoring of the photoreactive benzo(a)pyrene (BaP) (1 ng/m^3^ per year–standard) [22]. The pro-cancerogenic potency of air pollutant BaP could be accelerated by sunlight exposure [23]. Anthracene, BaP, and naphthalene are direct irritants and sensitizers of the skin.

O_3_ is a highly reactive pollution oxidant formed by the interaction of VOCs and gaseous NO compounds upon UV-photoactivation. Even though O_3_ is not a radical species per se, its noxious effects on the epidermis are mediated through the generation of potentially toxic peroxidation products and antioxidant depletion [24]. In fact, it is proven that exposure of human cutaneous tissue to O_3_ increases lipid peroxidation byproducts: α-β unsaturated aldehyde 4-hydroxynonenal (4HNE) and 8-iso-prostaglandin-F_2α_(8-iso F_2α_) [25].

The skin is also a target of another physical stressor represented by UV radiation (UVR). UVR is the most harmful environmental factor that affects skin biology contributing to photodamage on chronically sun-exposed areas, thus accelerating the physiological aging process and increasing skin cancer risk [26,27]. UVR wavelengths of solar radiation reaching the Earth’s surface include UVB (290–320 nm) and UVA (320–400 nm), each having a different mechanism of action on the skin and other tissues exposed to radiation. For example, the biological response to UVB is predominantly dependent on the absorption of its energy by chromophores such as DNA, pyrimidines and purines, aromatic amino acids, indoles, proteins, and biogenic amines, leading to tissue damage and genetic mutations, and to a lower degree, on the generation of reactive oxygen species (ROS). UVA biological activity, on the other hand, is primarily dependent on ROS and reactive nitrogen species (RNS) production with absorption by limited cellular chromophores such as NADH, NADPH, riboflavin, and porphyrins, and being weakly absorbed by DNA [26,28]. Both wavelengths have different levels of skin penetration. For example, UVB primarily penetrates the epidermis and, to a lower degree, the papillary dermis [26,29]. On the other hand, UVA penetrates deeply into the reticular dermis, also reaching the hypodermis [26,29]. UVR also influences the composition of the cutaneous microbiome [30]. Moreover, the co-exposure of air pollutants (PM/PAHs) and UVA, defined as photopollution, causes an additive damaging effect on the skin [31].

Until recently, the deleterious effects of sunlight on the skin were assigned to the UVR. Recently, studies have demonstrated the negative impact of the most energetic light from the visible spectra, blue light, on the skin and the circadian rhythm [32,33]. With the widespread use of electronic devices, the problem of light pollution is becoming increasingly serious for human health. Humans are exposed to blue light from electronic screen devices (456 nm) and light-emitting-diodes (LEDs) during both day- and nighttime in modern society. In comparison with UV rays, blue light has deeper penetration into the subcutaneous tissue [34]. There is convincing evidence for retinal damage and degeneration caused by blue light pollution through the induction of ferroptosis [35]. On the skin, blue light irradiation can cause changes in skin chromophores and can induce hyperpigmentation (in dark skin) as a visible sign of photoaging [36]. 

## 2. Molecular Mechanisms of Pollutant-Affected Skin Physiology

Air pollutants and photopollution exert a negative impact on the skin health of the general population. They compromise cutaneous structural integrity, interact with human skin microbiota, and trigger or exacerbate a range of skin diseases through various mechanisms [4,37,38], which are briefly discussed below.

### 2.1. Skin-A Dynamic Barrier with the Capacity to Maintain Homeostasis

The skin is a complex, multifunctional organ with the capacity to maintain cutaneous and overall body homeostasis, electrolyte and fluid balance, and thermoregulation [39,40,41]. It operates as a fully functional and independent neuroendocrine biofactory producing hormones, neuropeptides, and neurotransmitters [41,42,43,44,45]. On the other hand, the skin serves not only as a source of hormones but also as a target for their activities [39,41,46,47]. In addition, UVB can induce the production of the prohormone vitamin D, which then transforms into biologically active metabolites [48,49]. Of note, vitamin D also plays an important role in epidermal barrier functions and cutaneous homeostasis protection [50] and participates in the modulation of skin immunity [51]. Furthermore, skin possesses an interactive network between the nerves, neuroendocrine signals, and the cutaneous immune system (NICE), securing homeostasis and skin integrity [45,52,53,54,55,56,57,58,59].

Facing the environment, the skin with its epidermal barrier [60] protects the body from the deleterious insults of its surroundings. Environmental factors trigger cutaneous changes that can imprint immune cells acting as messengers of skin responses to regulate local and systemic homeostasis. Frequently, skin is exposed to a mixture of air pollutants. The impact of air pollutants on the skin might be different depending on their particular characteristics and intensity, and the length of their exposure. The underlying mechanisms of skin damage also differ based on cutaneous contact with the particular pollutant or co-pollutants. Generally, air pollution elicits an oxidative stress response in the skin with the activation of inflammation, which drives the skin and systemic pathologies. These effects on health could be amplified by the deleterious synergic effect of chronic UVR exposure [61]. Air pollution can induce or exacerbate various skin pathological conditions and systemic diseases (Figure 1) such as premature skin aging, skin cancer (cutaneous melanoma, squamous cellular carcinoma (SCC), and basal cell carcinoma (BCC), inflammatory skin diseases (atopic dermatitis (AD), airborne contact dermatitis (ABCD), allergic contact dermatitis, and psoriasis, as examples), acne, alopecia including androgenic alopecia, pigmentary disorders such as vitiligo, melasma, post-inflammatory pigment alteration, different neurodermatoses, itching disorders, etc. [3,38]. 

### 2.2. Cutaneous Response to Air Pollutants, Photopollution, and Cigarette Smoke

It is well established that the skin responses to air pollutants, cigarette smoke, and blue light exposure lead to an induction of oxidative stress [62,63]. Several air pollutants, especially those containing PAHs (e.g., PMs, cigarette smoke), which are highly lipophilic, alone or after UVA exposure, by generating ROS/RNS, can significantly impair cellular redox homeostasis [31,64,65]. UFPM, with their small size, could enter the mitochondria via cellular uptake and promote ROS production resulting in mitochondrial impairment from oxidative processes [66]. Moreover, PM-linked mitochondrial DNA (mtDNA) oxidative injury was reported in maternal and umbilical cord blood in women exposed to PMs in various time windows during their pregnancy [67]. Additionally, PMs exhibit oxidative potential from their metal constituents (Fe, Cu, etc.) able to catalyze Fenton-like reactions, triggering ROS-associated oxidative damages [68]. 

It is well-accepted that oxidative stress leads to lipid peroxidation, protein carbonylation, and genomic and mtDNA oxidative damage. The main product of pollution-driven oxidative stress is the highly reactive compound 4HNE. It results from the non-enzymatic peroxidation of ω-6 polyunsaturated fatty acids (PUFAs) in the biomembranes, such as linolenic acid, which is essential for the skin [69]. 4HNE can be produced directly upon O_3_ exposure by oxidizing PUFAs present in the upper cutaneous layers, forming unstable peroxides, and indirectly after PMs and cigarette smoke contamination. It is believed that PMs and cigarette smoke trigger 4HNE production via the generation of ROS/RNS [64,70]. Free radicals initiate the oxidation of arachidonic acid leading to the formation of isoprostanes F_2α_ (8-iso-F_2α_), which can increase phospholipid peroxidation and PUFAs oxidization while increasing the 4HNE concentration [71].

In addition, cutaneous exposure to O_3_ causes oxidative imbalances in the skin via the depletion of hydrophilic (vitamin C, glutathione, and uric acid) and lipophilic antioxidants (vitamin E) in the upper layers of the epidermis of human [72] and murine skin [73,74]. Of note, the stratum corneum has been identified as a primary target of O_3_-induced skin damage. The lipids on the skin surface derive from epidermal and sebaceous sources. Squalene is the major sebum-specific human skin lipid serving as a source of lipid peroxide on the skin surface [75]. As a defense against oxidative processes on the skin, vitamin E of nutritional origin is actively secreted from sebaceous glands and presumably co-secreted with squalene [76]. The main environmental stressors, such as O_3_, long UVA rays, PMs, and tobacco smoke, are able to boost the (per)oxidization process of squalene, resulting in the production of comedogenic molecules and the impairment of cutaneous physiology [77,78]. The biochemical changes in the sebum and corneal layer of the skin indicate pollutant-induced oxidative stress responses in humans living in urban areas, with decreased concentrations of squalene, vitamin E, and ATP and increased levels of carbonylated proteins [79]. In addition, a recent study showed an iron-dependent accumulation of lipid peroxides (ferroptosis) in female skin exposed to cigarette smoke [80].

### 2.3. AhR Signaling Pathway–A Target of Air Pollutants

The aryl hydrocarbon receptor (AhR) is expressed in all epidermal and dermal cells and is essential for skin integrity and skin immunity [81,82]. It is widely considered that AhR exerts a dual role in skin physiology and pathology, including the regulation of melanogenesis, inflammatory responses, and skin cancerogenesis [83]. AhR is a sensor for a variety of molecules and its ligands are abundant in the skin. Their nature (endogenous or exogenous sources) determines the phenotypic responses [82]. Solar UVB radiation generates the AhR endogenous ligand from trypthophan called 6-formylindolo[3,2-b]carbazole (FICZ) in the skin. FICZ promotes melanin pigmentation and improves DNA protection [84], similar to melanin [85]. AhR, being a sensor of environment–skin interactions, plays a crucial role in responses to (photo)pollution. Various air pollutants and tobacco smoke contain many synthetic exogenous ligands with very high affinity to AhR, which some used to call the “dioxin-receptor” [86]. PAHs (such as BaP) and dioxins from pollutants are capable of binding and triggering the AhR canonical signaling pathway. Once activated, AhR is translocated to the nucleus, where it dimerizes with the AhR nuclear transporter (ARNT). The protein complex AhR/ARNT binds to the xenobiotic-responsive element (XRE) in promoter regions and then induces the transcription of AhR-responsive genes, such as cytochrome P450 1A1 (*CYP1A1*) [87]. After the interaction with canonic XRE, AhR is transported back into the cytosol and its ligands are metabolized by CYP1A1. This action of CYP1A1 is linked to the production of a large number of ROS and mutagenic PAH metabolites, which may damage proteins and DNA. Therefore, the canonical pathway activated by AhR may be responsible for premature skin aging, skin cancer, and the inhibition of apoptosis. AhR also promotes the transcription of its own AhR repressor (AhRR) allowing negative feedback regulation [88]. On the other hand, there are opposite data and concepts showing that the activation of AhR can induce anti-oxidative and anti-inflammatory responses and that it can regulate epidermal barrier functions and exhibit anticancerogenic activity [89,90,91,92]. Furthermore, there is a growing body of evidence that the AhR is also a receptor for several compounds such as vitamin D3 hydroxyderivatives [93,94,95], and diverse beneficial effects of active forms of vitamin D are widely recognized.

The altered redox homeostasis from air pollution initiates downstream signaling cascades interplaying between inflammation and oxidative stress products, thus contributing to a vicious circle of skin oxinflammation. Depending on the ligand and context, AhR can generate an oxidative stress response, enhanced expression of pro-inflammatory genes, and apoptotic cell death [96,97,98]. Pollutant-activated AhR can trigger the production of inflammatory and immunosuppressive cytokines in cutaneous keratinocytes and fibroblasts. In vitro studies have demonstrated that airborne fine particles dose-dependently enhanced the release of interleukin-6 (IL-6), IL-8, IL-1α, and IL-1β in the skin [64,65,70,97]. Similarly, BaP from cigarette smoke can induce oxidative stress-associated IL-1α in adults [99] and IL-8 production in human epidermal keratinocytes via the AHR signaling pathway [100]. This study supports the clinical observations that smoking exacerbates IL-8-related inflammatory skin diseases, such as acne and palmoplantar pustulosis [101,102].

PMs and O_3_ can cause the upregulation of cyclooxygenase-2 (COX-2) expression and the activation of nuclear factor kappa beta (NF-κβ) in keratinocytes and fibroblasts [97,98,103]. The redox-sensitive transcription factor NF-κβ might be additionally connected to a noncanonical AhR inflammatory pathway (interacts directly with AhR in the absence of AhR/ARNT activity) [104]. After binding to AhR, urban PMs can stimulate the phosphorylation of extracellular signal-regulated kinases (ERK)/p38 mitogen-activated protein kinases (MAPKs) and transcription factor c-Jun-N-terminal kinase (JNK) signaling via the nicotinamide adenine dinucleotide phosphate (NADPH)-oxidase (NOX)-generated ROS. It was proposed that NOX/ROS-dependent mechanisms could induce NF-κβ activation and phosphorylation of activator protein-1 (AP-1) [105]. After PM exposure, the phosphorylated dimers of the AP-1 transcription factor, c-Jun, and c-Fos, as well as the p38 kinase, regulate the transcription of *IL-1α* and *IL-1β* genes in human keratinocytes [97,105]. Moreover, the activation of p65 (a subunit of NF-κβ) and c-Jun/c-Fos contribute to a significant upregulation of COX-2 expression and PGE_2_ production induced by PMs [98,105]. In addition to COX-2, IL-1α and IL-1β are capable of regulating matrix metalloproteinase-1 (MMP-1) expression. In an in vitro co-culture model of human keratinocytes and fibroblasts, it was assumed that after PM treatment, keratinocyte-derived IL-1α/1β would increase the levels of MMP-1 and COX-2 in the fibroblasts [97]. Likewise, tobacco smoke extracts can induce *MMP-1* mRNA expression in vitro in human skin cells in a dose-dependent manner [106]. Moreover, using a 3D skin model, consisting of an epidermis without a dermis, Lecas S. et al. (2016) demonstrated the noxious effect of tobacco smoke (containing both PM and gas) that resulted in the significant production of inflammatory cytokines (IL-8, IL-1α, IL-18) and MMPs (MMP-1 and MMP-3), as well as an increase in the 4HNE level [107]. The activation of ERK and MAPK p38 from air pollutants and cigarette smoke favor inflammation-induced degradation of the extracellular matrix (ECM) and downregulation of neocollagenesis. MMPs can fully degrade collagen and elastin fibers, thus decreasing the skin’s viscoelasticity. MMP-1 (collagenase) is the major collagenolytic enzyme that contributes to the breakdown of ECM of the skin tissue, primarily initiating the fragmentation of type I and III collagens. Once cleaved by MMP-1, collagen can be further degraded by both MMP-3 and MMP-9 [108]. At the same time, cigarette smoke and air pollutants impair the biosynthesis of collagen types III, IV, and VII, as well as induce the neosynthesis of abnormal elastin fibers [106]. These alterations are responsible for wrinkle formation and premature skin aging [109,110].

The AhR can also contribute to the modulation of growth factors such as epidermal growth factor (EGF) and transforming growth factor-β (TGF-β). EGF is involved in cell differentiation and proliferation. PAHs-induced AhR activation can cause a decrease in skin cell proliferation together with the apoptosis of keratinocytes and fibroblasts [111]. In addition, the treatment of keratinocytes with 2,3,7,8-tetrachlorodibenzo-*p*-dioxin (TCDD) significantly accelerates the rate of terminal differentiation and cornification [112,113]. EGF through the EGF receptor (EGFR) mediates the opposing effects on gene transcription and cell differentiation in comparison to the TCDD-induced AhR activation in human keratinocytes [112]. On the other hand, active TGF-β plays an essential role in epidermal homeostasis and dermal growth. PAHs, including those in cigarette smoke, have induced the non-functional latent form of TGF-β and the downregulation of the TGF-β1 receptor. These data are consistent with the observations of smoking-associated premature skin aging through the impairment of ECM protein synthesis [114].

Environmental air pollutants exert an important impact on skin quality. Airborne PMs, O_3_, smoking, VOCs, NO_2_, and UVR are capable of increasing trans-epidermal water loss (TEWL) and compromising the epidermal barrier function and structural integrity of the skin, which in turn induce or exacerbate various skin diseases [115,116]. In keratinocytes, PMs downregulate the expression of genes of terminal differentiation markers such as filaggrin (*FLG*) by their COX-2/PGE_2_ oxidative potential [105]. Filaggrin contributes to the formation of the corneal layer, thus playing an essential role in the barrier function of the skin. Additionally, a tobacco smoke-exposed reconstructed epidermis model showed even slight exposure triggered drastically decreased expression of both types of epidermal structural genes, those encoding cell–cell adhesion (desmoglein 1 (*DSG1*), desmocollin 1 (*DSC1*), and corneodesmosin (*CDSN*)) and terminal differentiation genes (filaggrin (*FLG*), loricrin (*LOR*), involucrin (*IVL*), etc.), and their corresponding proteins. Consequently, skin architecture can be modified by cigarette smoke, exhibiting poor cohesion of the epidermal barrier and accelerated desquamation and cornification [107]. The crucial role of the AhR signaling pathway for skin barrier integrity can be seen in AhR-knockout mice, which display weaker intercellular connectivity in keratinocytes and express lower expression levels of genes associated with the skin barrier function [117].

While we have focused on pollutant-induced AhR signaling pathways’ involvement in epidermal barrier defects, alteration in epidermal cell differentiation and proliferation, hypopigmentation, and the increased production of proinflammatory cytokines, we must also emphasize the opposite effects, e.g., building the epidermal barrier, anti-inflammatory, antioxidative, and anticancer activities, and defining the Yin and Yang mechanism of action of the AhR, which will be dependent on the nature of the ligand and the cellular and biochemical context (Figure 2).

### 2.4. Pollution Affects Vitamin D Production in the Skin

Air photo(pollution), environmental chemicals, and smoking can be important factors for vitamin D deficiency [118]. PMs and O_3_ can directly affect the cutaneous production of vitamin D [118]. In addition, air pollutants, persistent organic pollutants, and heavy metals can behave as endocrine-disrupting chemicals (EDCs), which may alter vitamin D-hydroxyderivatives levels [119]. Other environmental factors, such as the winter season, inadequate sun exposure, and a high-latitude location, can also predispose one to vitamin D deficiency [120,121]. Exposure to UVB rays accounts for more than 90% of vitamin D production in humans [48,122,123]. Air pollution also affects vitamin D synthesis by reducing UVB energy reaching the surface of the Earth, which is essential for the production of vitamin D_3_, tachysterol (T_3_), and lumisterol (L_3_) derivatives [49,124,125,126]. It should be noted that exposure to air pollutants during the gestational period can decrease vitamin D levels in newborns [127]. The absorption/scattering of UVB radiation caused by tropospheric ozone and PMs is becoming a real concern for human health [128]. The skin serves not only as a primary place for vitamin D_3_ production but also as a target organ for its action, which is vital in the formation of the skin barrier and hair follicles [125]. Moreover, vitamin D deficiency has been connected to many inflammatory, proliferative, and malignant skin disorders [129,130,131,132]. There is growing experimental evidence demonstrating the photoprotective properties of vitamin D_3_ and L_3_ hydroxyderivatives [133,134,135,136,137,138,139]. The classical and novel forms of vitamin D also impact the development and progression of melanoma and non-melanoma skin cancers [140,141,142,143,144,145].

## 3. Pollution-Induced Skin Pathologies

### 3.1. Premature Skin Aging

Skin aging is a process driven by the total exposure of both intrinsic and extrinsic factors over the human lifespan (skin exposome), which is responsible for progressive morphological and functional cutaneous alterations [146,147,148]. The most prominent external stressors for the skin, which can result in premature aging, include UVR [149,150], air pollutants [110,151,152], and smoking [153]. They predominantly affect parts exposed to the environment such as the face, head, neck, and back of the hands. Moreover, chronic photopollution exposure of the skin may accelerate the photoaging process and can correlate with enhanced cancer risk [31,154].

The endogenous antioxidant cutaneous capacity decreases with aging, making aged skin more vulnerable to environmental insults. Indeed, human skin aging is primarily induced by oxidative events. In particular, extensive ROS production and insufficient scavenging activity or mitochondrial dysfunction are crucial for premature skin aging [155,156,157]. O_3_ from smog and PM can trigger ROS generation and induce oxidative stress, contributing to phenotypic features of aging. O_3_ primarily causes wrinkles formation [148,158], while long-term exposure to fine PM and NO_2_ is responsible for pigment spots (lentigines) [159] and deep nasolabial folds [160]. Currently, the habit of smoking is considered an aging accelerator, and it is an even greater contributor to facial wrinkling than sun exposure [161,162]. Smoking also increases keratinocyte dysplasia and skin roughness and increases the risk of the development of pathological elastosis [153]. Moreover, long-term secondhand-smoke exposure could also affect skin appearance as shown in a premature skin aging rat model [163].

Cigarette smoke and PMs contain numerous PAHs that trigger premature skin aging via an AhR pathway [164]. Activated AhR can subsequently initiate ERK/MAPK signaling pathways and upregulate the expression of aging-related genes [111]. MMP-1, which degrades collagen, is induced dose-dependently by tobacco smoke extracts in human fibroblasts and keratinocytes through AhR activation [106]. Similarly, in primary human dermal fibroblasts exposed to different concentrations of PMs, the expression of MMP-1, MMP-3, and TGF-β significantly increased and showed changes associated with skin aging [165]. Indeed, senescent fibroblasts are characterized by increased levels of MMP secretion. In addition, PM_10_ may contribute to premature skin aging and skin inflammation via impaired collagen synthesis [166].

### 3.2. Skin Cancers

Solar UVR is the primary etiological factor responsible for skin cancer development. It is found that SCC and BCC are dependent on the total dose of UVR exposures, while cutaneous melanoma is also dependent on exposure patterns, with intermittent irradiation being the most harmful [167]. During the last few decades, humans worldwide have been facing the adverse influence of climate change and stratospheric ozone depletion, which are becoming a public health concern [168]. The changes to the O_3_ layer resulted from global industrialization and ambient pollution, which allow an increased proportion of UVB irradiation to reach the Earth’s surface. This depletion of the O_3_ layer can impact the dual effects of UVR on skin health [26,169] and likely enhance the risk of skin cancer. Every 1% of O_3_ layer reduction would increase UVB irradiation by 2%, which may result in a 1 to 2% increase in cutaneous melanoma and a 3 to 4.6% increase in SCC incidence [170]. Nonmelanoma skin cancer (SCCs and BCCs) is the most common type of human cancer, primarily affecting white-skinned individuals. Importantly, the incidences of both melanoma and non-melanoma skin tumors are increasing worldwide [171].

Ambient air pollution alone or simultaneously with UVR may play a crucial role in skin cancer emergence. Many air pollutants are potentially cancerogenic. Among these pollutants, PM_10_ is believed to be linked to enhanced cancer development of the skin, larynx, lungs, thyroid gland, and bladder. Moreover, it was established that a 10 µg/m^3^ increase in the PM_10_ concentration was associated with a 52% higher relative risk of nonmelanoma skin tumors [172]. Moreover, UFPMs including black carbon and PAHs increase the incidence of skin cancer [173]. BaP and other PAHs are photoreactive, and their potential cancerogenic risk seems to be aggravated by UVA radiation [21,174]. In addition, the higher risk of developing SCCs in smokers [175] is mostly attributed to the carcinogen BaP, present as one of the main constituents of cigarette smoke [176]. In the presence of PUFAs (e.g., in biomembranes), BaP undergoes oxidative transformations and the products of BaP oxidation and lipid peroxidation exhibit mutagenic potential and may induce cancerogenesis [177,178].

Chronic exposure of the skin to PAHs and/or UVR may activate the AhR transcription factor and promote downstream epidermal targets, contributing to skin cancerogenesis [179,180]. Other AhR ligands have the potential to increase PAH toxicity through the induction of the *CYP1A1* gene. The interaction of pollutants with AhR can induce pro-oxidative mechanisms and oxidative stress-induced genotoxicity [181]. Enhanced ROS formation and reduced antioxidant defense can be correlated with DNA damage, mutations, and cancerogenesis. The critical role of AhR in PM- or BaP-induced cancerogenesis has been indicated by the lack of skin tumors in AhR^(−/−)^ mice continuously exposed to airborne PM or BaP. In this setting, tumor development in AhR^(+/+)^ mice was mediated via *CYP1A1* expression induced by PAHs [182,183]. In addition, mice with Langerhans cells (LCs) deletion were protected from cancerogenesis, independent of T cell immunity. LC-deficient mice skin has shown relatively strong resistance to cancer development. LCs, which are skin-specific dendritic cells, metabolize PAHs into a pro-oncogenic intermediate, which increases mutagenesis and facilitates epidermal DNA damage and SCC formation [184].

### 3.3. Atopic Dermatitis

Atopic dermatitis (AD) is a chronic inflammatory cutaneous disease affecting up to 20% of children and 10% of adults, which is growing globally with prevalence in industrialized countries [185,186]. This disorder has a complex pathogenesis that relies on environmental and genetic factors, epidermal barrier dysfunction, immune dysregulation, cutaneous and gut dysbiosis, and food sensitization [187,188]. Air pollutants, especially those due to industrialization and wildfires, could be independent factors related to increased risk for the development or exacerbation of AD lesions and symptoms in both children and adults [189,190,191]. Moreover, prenatal exposure to PMs (PM_2.5_ and PM_10_) and NO_2_, especially during the first trimester, was linked to a higher risk of infantile AD [192,193,194]. Moreover, smoking, both active and passive, positively correlates with adult AD prevalence [195,196,197]. Several studies have demonstrated that NO_2_, VOCs, and formaldehyde can increase trans-epidermal water loss (TEWL), which may result in skin barrier damage [198,199,200] and hypersensitization to aeroallergens leading to systemic allergic responses [201,202]. On the other hand, epidermal barrier disruption, especially in AD children’s skin, allows the intracellular penetration of PMs. Repeated PM exposure can cause thickening of the epidermis and dermal inflammation with neutrophil infiltration [203]. Additionally, it was established that the effects of atmospheric particles of various sizes and gaseous air pollutants (e.g., NO_2_, SO_2_) on atopic skin could be augmented by meteorological factors, especially dry conditions and high temperatures [204,205]. Furthermore, environmental stressors, including airborne pollutants, may change the skin microbiome composition, thus influencing the epidemiology and severity of AD [206]. It was well documented that AD lesional skin with abnormal cutaneous barrier possesses abundant colonization of *Staphylococcus aureus*, *Corynebacterium*, etc. species, which further exacerbate T helper 2 (Th2)-deviated skin inflammation [207].

In vivo and in vitro studies showed that air pollutants contribute to the development and flares of lesions and symptoms of AD via the activation of AhR signaling pathways, promoting oxidative stress, triggering a proinflammatory response, and impairing the skin’s barrier function [208,209]. For example, in an AD mouse model, PM_10_ application induced/aggravated cutaneous inflammation through differential expression of genes encoding skin barrier integrity and immune responses [210]. At the molecular level, it was found that AD lesions exhibited an elevated expression of AhR and ARNT [211]. Furthermore, the levels of AhR in the serum of patients with AD increased compared to healthy individuals. AhR mRNA expression positively correlates with the severity index score of the disorder and the concentration of IL-6 in the serum of AD patients [212]. Cigarette smoke can additionally affect atopic sensitization via the BaP–AhR interaction. This interaction further causes immune responses including T-cell polarization and LCs migration [213]. In chronically inflamed skin such as in AD, increased levels of noncanonical AhR-partner molecules, such as NF-κB and the signal transducer and activator of transcription-3 (STAT-3), are noted [104].

The activation of AhR in human keratinocytes by air pollutants can induce elevated expression of the neurotrophic factor artemin (ARTN) in the epidermis of patients with AD but not in healthy people [214]. Recently, Hidaka et al. also studied the expression of *Arnt* in mice expressing constitutively active AHR in keratinocytes (AHR-CA). Those AhR-CA mice overexpress *Artn*, which, as a target gene of AhR, can induce substantial pruritus during the chronic presence of PAHs, as evaluated by continuous scratching behavior [214].

### 3.4. Psoriasis

Psoriasis is a common (2–3% worldwide prevalence) chronic inflammatory skin disease with systemic involvement. As an immune-mediated disorder, Th17 cells and IL-17 play a crucial role in its pathogenesis [215]. It is thought that short-term exposure to ambient air pollution may be a trigger factor for psoriasis flares [216,217]. Moreover, cigarette smoke may increase the severity of psoriasis in a dose-dependent manner [218]. Airborne pollutants such as PMs [219] and O_3_ [220] may play an essential role in Th17 differentiation in an AhR-dependent manner [220,221]. The AhR and ARNT are found co-localized at the lower layer of the epidermis in acute psoriatic lesions, which also suggests the activation of AhR pathways [205]. Moreover, modulated by AhR, autophagy can lead to skin inflammation in human keratinocytes via the p65NF-κβ/p38MAPK signaling pathways, suggesting that AhR signaling and autophagy may contribute to psoriasis-related inflammation [222].

### 3.5. Acne

Acne vulgaris is a common inflammatory skin disease involving the pilosebaceous unit, with multifactorial etiology where an exposome [223] along with increased sebum production due to hormonal stimulation and genetic factors are very important etiologic factors [224,225]. Ambient pollution is considered one of the exposome factors of the skin. Acne could be either aggravated or developed in highly polluted areas. A relationship between air pollutants and increased acne prevalence has been established [226]. In studies with Chinese subjects, it was found that cutaneous exposure to increased levels of PMs (PM_2.5_ and PM_10_) and NO_2_ were significantly associated with a higher number of acne lesions and increased sebum production (hyperseborrhoea) [226] that resulted in more outpatient visits [227]. Notably, increased concentrations of SO_2_ showed a negative correlation with the number of acne-related daily visits [227]. This is in accordance with the use of sulfur-containing topical combinations for the treatment of acne vulgaris. Another study demonstrated age- and gender-related effects of short-term exposure to air pollutants on the skin and the risk of increased outpatient visits for teenage and adult acne. [228]. Both SO_2_ and NO_2_ were found positively correlated with acne presentation in men and women. SO_2_ was found to be associated with increased acne visits only in children and young adults in this study, whereas NO_2_ corresponded to increased acne visits in all age groups [228]. In addition, types of food could affect acne presentation.

The exact mechanism of acne formation in the presence of air pollutants is still not fully clarified. As we discussed above (Section 2.2), air pollution inducing oxidative stress in the skin can cause lipid peroxidation, the carbonylation of proteins, and oxidative DNA damage. Human sebum produced by sebaceous glands during the life cycle of sebocytes is a mixture of triglycerides, wax esters, and squalene. Sebum secretion fuels the lipophilic resident cutaneous flora, primarily comprising *Propionibacterium acnes*, *Staphylococci*, and *Malassezia* species [229]. O_3_, cigarette smoke, and long UVA rays exhibit the powerful (photo)oxidizing potential of squalene [24,78]. In particular, the overlying lipids on the corneal layer of the epidermis can be modified by air pollutants with increased levels of peroxidized forms of squalene and decreased levels of linoleic acid, thus promoting comedogenesis [78,230]. Moreover, hyperseborrhea and dysseborrhea, which are important etiopathogenetic factors in acne [231], were measured on the forehead of citizens in two highly polluted cities, namely Shanghai and Mexico City [79,232].

Outdoor air pollution could also trigger inflammation of the skin and stimulate the occurrence of inflammatory acne lesions [233]. PAH-activated AhR may further induce skin inflammation via the heightened production of pro-inflammatory cytokines from epidermal keratinocytes such as IL-1α [97,99] and IL-8 [100]. Both cytokines are more prominent in inflamed acne lesions than in normal skin, demonstrating their contribution to the exacerbation of acne [99,230,234]. The colonization of *Propionibacterium acnes* in areas of the skin with hyperseborrhea additionally drives inflammatory responses in acne vulgaris [235].

In addition, exposure to a high concentration of dioxins via inhalation, ingestion, and transcutaneous absorption induces chloracne, which is classified as an occupational dermatosis [236]. Dioxins have been found to be very potent ligands of AhR and promote accelerated terminal differentiation and hyperkeratinization of the epidermis, and convert sebocytes towards a keratinocytic differentiation lineage, which can result in acneiform eruptions [237].

### 3.6. Melasma and Other Hyperpigmentations

Melanogenesis and pigmentation are vital in photoprotective responses against DNA damage in epidermal keratinocytes [238]. In fact, photoprotection by tanning can occur even without UVB exposure [239].

AhR is a pleiotropic sensor of environmental factors. UVB-induced, FICZ-mediated activation of AhR is essential for the protection of DNA. Moreover, it was confirmed that AHR-deficient mice exhibit significantly weaker skin tanning in comparison with wild-type mice [84]. The activation of AhR/MAPK signaling in melanocytes in response to other ligands present in air pollutants (PAH, dioxins) promotes the proliferation of melanocytes and stimulates melanogenesis by upregulating the expression of melanogenic enzymes, which leads to hyperpigmentation and the appearance of pigmented spots on the skin [237,240,241].

### 3.7. Hair Loss

Airborne pollutants such as PM_s_, VOCs, NOx, and carbon oxides (CO and CO_2_) may be deposited on the hair and scalp or penetrate into the deeper layer of the skin via hair follicles or transcutaneous absorption [242]. Urban PM nanoparticles were observed inside hair follicles in both intact and barrier-disrupted skin in vivo [203]. Absorbing PAH on their surface, PMs from air pollution may trigger ROS generation and ROS-dependent inflammatory responses in the skin, which may further result in hair shedding [242]. Other data demonstrate that PMs may induce keratinocytes’ apoptosis in ex vivo cultured scalp hair follicles that could impair hair growth [243]. In addition, tobacco smoke containing PAHs is recognized as an oxidizing agent and is implicated in early-onset androgenetic alopecia in men [244]. Similarly, in an animal model, most C57BL/6 mice exposed to cigarette smoke for 3 months developed areas of alopecia and grey hair. The smoke-treated mice skin showed atrophic skin, reduced thickness of the subcutaneous tissue, a scarcity of hair follicles, and massive apoptosis of hair bulb cells at the alopecia edges [245].

## 4. Preventive and Treatment Options for Air Pollutant-Damaged Skin

### 4.1. Anti-Pollution Skin Care Strategies

The noxious effects of pollution on the skin with potentially systemic implications stirred interest in developing effective anti-pollution skincare and haircare products [246]. The main preventive and therapeutic strategies against air pollution-driven skin damage should include control of the deposition and penetration of air pollutants through the skin and hair follicles and their removal, repairing skin barrier function and improvement of hydration, control of pigmentation, reduction of oxinflammation by using antioxidants, and the prevention of collagen and elastin degradation [3,11].

The basic pollution protective measures involve the use of protective clothing, rinsing products to wash away the pollutants from the skin (but not overwashing), and barrier creams to preserve or restore the skin barrier [246,247]. Sun protection is essential for the prevention of photo-reactive compounds to cause photopollution skin damage. Therefore, products with broad-spectrum (UVA + UVB) sun protection, sunscreens with the addition of antioxidants (vitamin E, ferulic acid, etc.), and sunscreens and after-sun lotions containing DNA repair enzymes are a promising strategy against cutaneous photodamage [248,249,250]. Additionally, sunscreen with mineral filters containing pigments such as iron oxide and titanium dioxide could be helpful for melasma prevention [251].

Plants extracts and other natural compounds with powerful antioxidant activity can protect and treat air-pollution-related oxidative damage to the skin [63,252,253]. Of note, a comprehensive topical antioxidant mixture (WEL-DS) that combines 19 water-soluble, lipid-soluble, and enzymatic antioxidants, designed to protect skin from the oxidizing effects of UVR and to reduce the visible signs of facial photodamage [254], has diminished oxidative stress promoted by blue light and cigarette smoke exposure [63], as well as the ambient O_3_-induced oxidative damage in a reconstructed skin model [255].

Another potential candidate for the attenuation or treatment of (photo)pollution-triggered cutaneous damage is melatonin [256,257]. Melatonin, being both synthesized and metabolized in the skin, together with most of its metabolites, works as a potent antioxidant that possesses the capacity to reduce oxidative damage, protect mitochondrial functions, reduce inflammation, and modulate the immune system in the skin [42,258,259,260,261]. It should be noted that melatonin’s antioxidant cascade directly detoxifies many harmful radicals (hydroxyl radical and peroxynitrite), which are not degraded by enzymes in comparison with a single radical scavenge by classical antioxidants (vitamins C and E) [262,263,264]. Melatonin may additionally activate antioxidant enzymes, which are able to degrade the weakly reactive ROS [265,266]. Additionally, melatonin is capable of stimulating SIRT1, which plays an important role in pollution-associated premature skin aging. Thus, the upregulation of SIRT1 could downregulate MMP-1 and MMP-3, which contribute to collagen breakdown and could decrease inflammation via NF-κβ signaling [267]. The (photo)protective potential of topically applied exogenous melatonin has been demonstrated by many studies. For example, the application of night cream containing melatonin, carnosine, and extracts of the Mediterranean flowering plant Helichrysum italicum on skin explants pre-exposed to a mixture of PAHs and heavy metals showed a reduction of the cutaneous irritation and damage, and a significant increase in collagen type I [268]. Moreover, melatonin could reverse the increased lipid peroxidation and might even improve the skin of former smokers [269]. Therefore, formulating new skincare products with melatonin would be a novel approach to the prevention of oxidative damage on the skin caused by air pollutants, heavy metals, and smoking.

### 4.2. Modulation of AhR in Air Pollution-Induced Skin Diseases

Only a few specific molecules are currently able to prevent or counteract the noxious effects of ambient air pollutants on the skin. The use of AhR modulators has become a new strategy for anti-pollution skin prevention and therapy. Importantly, some AhR ligands can simultaneously activate the master regulator of the antioxidant responses, nuclear factor erythroid 2-related factor 2 (Nrf2) [96,104]. Other antioxidant phytochemicals (e.g., cinnamon derivatives) could activate Nrf2 and independently block the AhR and inhibit CYP1A1 [237,270,271]. The targeting of AhR/Nrf2 could be a potential candidate strategy in the treatment of air pollution-induced skin diseases. Therapeutics that both downregulate ligand-activated AhR signaling and stimulate the Nrf2 antioxidant system are likely the most promising candidates. The stimulation of the Nrf2 transcription factor further triggers the downstream antioxidative signaling pathway, neutralizing oxidative stress and playing a pivotal role in the cytoprotection of the skin [272,273,274].

Additionally, antioxidative AhR/Nrf2 dual agonists can reduce inflammation via the downregulation of proinflammatory cytokines and can regulate the skin barrier protein expression, improving epidermal barrier function [275,276]. Therefore, AhR/Nrf2 represents a potential therapeutic target in the treatment of inflammatory skin disorders such as atopic dermatitis, psoriasis, and other dermatoses [277,278].

Tapinarof is a novel, naturally derived, small-molecule AhR-modulating therapeutic agent, currently under clinical trials for the treatment of atopic dermatitis and psoriasis [279,280,281]. It specifically binds to and activates AhR in the skin, which inhibits cutaneous inflammation in mice and humans. The topical application of tapinarof cream on imiquimod-induced psoriasiform skin lesions in mice has led to a decrease in the cytokines level, as well as a reduction in epidermal thickening and erythema. In contrast, tapinarof has shown no effect on skin inflammation in AhR-deficient mice [282]. In addition, the significant efficacy of tapinarof in psoriasis is attributed to its downregulation of IL-17 and other pro-inflammatory cytokines. Tapinarof also exhibits antioxidative capacity via the activation of Nrf2 and its downstream antioxidative enzymes and downregulates the phosphorylated signal transducer and the activator of transcription 6 (P-STAT6) formation [283].

Topical tapinarof is a safe and promising agent with the ability to attenuate the development of atopic dermatitis and may also be helpful in the treatment of air-pollution-initiated exacerbation of atopic lesions. Tapinarof-treated NHEKs showed the upregulation of epidermal barrier *FLG* and *LOR* genes and protein expression via AHR activation. Tapinarof also enhanced the secretion of IL-24, a cytokine-activating Janus kinase (JAK)- STAT axis, which downregulates the expression of FLG and LOR [284]. It should be noted that IL-24 can be stimulated by certain oxidative AhR agonists such as TCDD, BaP, PMs, and UVB radiation [285].

## 5. Conclusions

The increased level of air pollution worldwide represents an environmental risk factor for skin damage and the development of cutaneous pathologies. Some PAHs are photoreactive and could be activated upon UVA light. Therefore, such UVR exposure would enhance their deleterious effects on the skin.

AhR is a sensor for small molecules such as air pollutants, which are capable of activating oxidative stress responses and triggering inflammatory and immunosuppressive cytokine production in the skin. On the other hand, targeting AhR/Nrf2 is emerging as a novel treatment option for ambient pollutants that induce or exacerbate inflammatory skin diseases such as atopic dermatitis, psoriasis, etc. Therefore, AhR with downstream regulatory pathways would represent a crucial signaling system regulating the skin phenotype in a Yin-and-Yang fashion defined by the chemical nature of the activating factor and the cellular context.

## Figures and Tables

**Figure 1 ijms-24-10502-f001:**
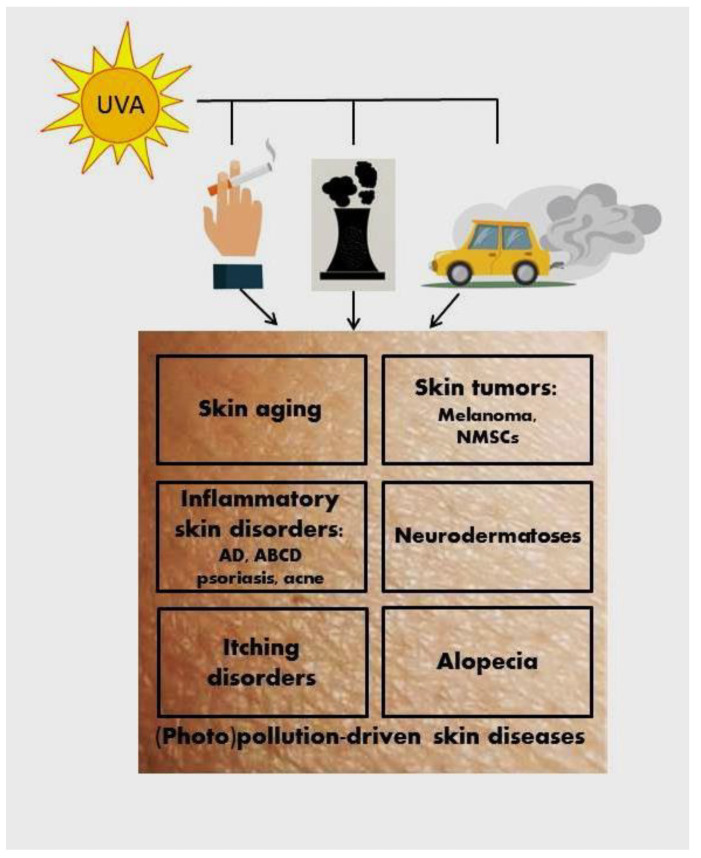
The deleterious synergic effect on the skin of UVA and air pollution. Non-melanoma skin cancer (NMSC); atopic dermatitis (AD); airborne contact dermatitis (ABCD).

**Figure 2 ijms-24-10502-f002:**
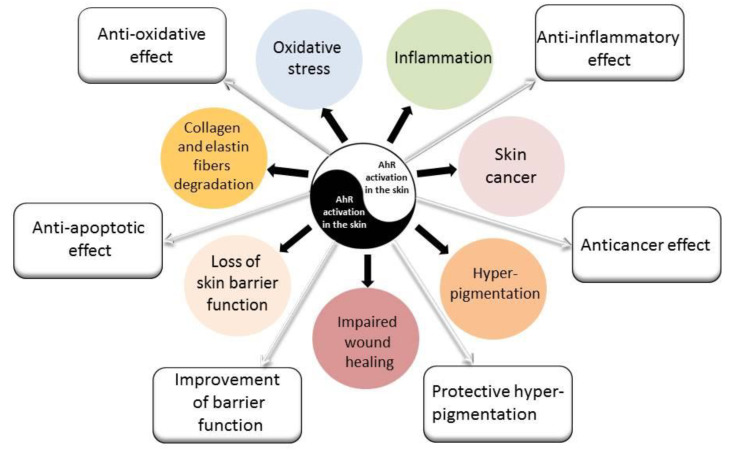
AhR signaling pathways’ activation and the cutaneous response to air pollutants and smoking. Negative (Yin) and positive (Yang) effects of AhR activation on the skin are marked by black and white arrows, respectively.

## Data Availability

All data appear in the manuscript. For further inquiries, please contact the first author or corresponding author.

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
