# Peer review of "Environmental Air Pollutants Affecting Skin Functions with Systemic Implications"

_ijms, 2023, doi:10.3390/ijms241310502_

Round 1

Reviewer 1 Report

Good topic.

No coherence between two paragraphs. Authors abruptly started a new paragraph with a new topic.

Authors did not provide any search criteria.

Title is very wide with an umbrella term "Environmental pollutants" but they did not discuss all relevant environmental pollutants and their association with skin disease. Either they change the title or add all relevant environmental pollutants in the manuscript.

English language is ok, can be improved. 

Author Response

The manuscript with revised title “Selective review of environmental air pollutants affecting skin functions with systemic implications” has been revised following reviewers suggestions and recommendations.

     We greatly appreciate the reviewers’ critique that has improved final presentation of the manuscript.

     Please, see our answers below for details.

Answer to the reviewer 1:

  1. No coherence between two paragraphs. Authors abruptly started a new paragraph with a new topic.

Reply

This has been corrected in the revised manuscript.

  1. Authors did not provide any search criteria.

Reply

Selection of the literature was based on the knowledge and experience of the authors who are experts in this field. The authors had also utilized Pubmed, and has searched using the following key words: environmental pollutants, skin, skin pathology, skin allergy, ultraviolet radiation and AhR. And finally, the selection of literature was further enhanced by utilizing AI tools such as ChatGPT-4.

  1. Title is very wide with an umbrella term "Environmental pollutants" but they did not discuss all relevant environmental pollutants and their association with skin disease. Either they change the title or add all relevant environmental pollutants in the manuscript.

Reply

In response to the critique, the title has been modified.

  1. English language is ok, can be improved.

Final proof-reading was done by Dr Radomir Slominski who is a native English (American English) speaker.

Reviewer 2 Report

This review entitled “Environmental pollutants affecting skin functions with systemic implications”. The authors have fully explained and cited relevant literature on the hazards of air pollution to human health, and the skin's exposure to a mixture of harmful air pollutants affects its physiological functions and damage mechanisms. This article has a very high academic and clinical diagnosis and treatment reference value.

 But there are still some things that can be further strengthened and supplemented:

1. In addition to the impact of air pollution on skin health, it can be briefly added that food can also affect skin health, such as Atopic dermatitis (AD) and Psoriasis.

2. The color vividness in Figure 1 can be further improved.

3. The author can attach a skin composition figure, and UVR effect penetration depth map to understand the effect of UVB on the skin's synthesis of vitamin D3 and its damage to the skin.

Author Response

     The manuscript with revised title “Selective review of environmental air pollutants affecting skin functions with systemic implications” has been revised following reviewers suggestions and recommendations.

     We greatly appreciate the reviewers’ critique that has improved final presentation of the manuscript.

     Please, see our answers below for details.

Answer to the reviewer 2:

The authors have fully explained and cited relevant literature on the hazards of air pollution to human health, and the skin's exposure to a mixture of harmful air pollutants affects its physiological functions and damage mechanisms. This article has a very high academic and clinical diagnosis and treatment reference value.

 But there are still some things that can be further strengthened and supplemented:

  1. In addition to the impact of air pollution on skin health, it can be briefly added that food can also affect skin health, such as Atopic dermatitis (AD) and Psoriasis.

Reply

We thank for the critique and have added the information that food can also affect skin health, and can aggravate inflammatory disorders such as  atopic dermatitis (AD) and psoriasis

  1. The color vividness in Figure 1 can be further improved.

Reply

This has been corrected as requested by the reviewer.

  1. The author can attach a skin composition figure, and UVR effect penetration depth map to understand the effect of UVB on the skin’s synthesis of vitamin D3 and its damage to the skin.

Reply

We thank the reviewer for the suggestion. The senior author has published large number of such figures in several high impact factor reviews and book chapters. To avoid redundancy in graphical presentation we now refer to such papers. We also discuss penetration of UVR into the human skin, especially UVB.

Round 2

Reviewer 1 Report

Well done!